# Secretory Vesicles Are the Principal Means of SARS-CoV-2 Egress

**DOI:** 10.3390/cells10082047

**Published:** 2021-08-10

**Authors:** Sébastien Eymieux, Rustem Uzbekov, Yves Rouillé, Emmanuelle Blanchard, Christophe Hourioux, Jean Dubuisson, Sandrine Belouzard, Philippe Roingeard

**Affiliations:** 1INSERM U1259 MAVIVH, Université de Tours and CHRU de Tours, 37032 Tours, France; sebastien.eymieux@univ-tours.fr (S.E.); emmanuelle.blanchard@univ-tours.fr (E.B.); christophe.hourioux@univ-tours.fr (C.H.); 2Plate-Forme IBiSA de Microscopie Electronique, Université de Tours and CHRU de Tours, 37032 Tours, France; rustem.uzbekov@univ-tours.fr; 3Faculty of Bioengineering and Bioinformatics, Moscow State University, 119992 Moscow, Russia; 4U1019-UMR 9017-CIIL-Center for Infection and Immunity of Lille, Institut Pasteur de Lille, Université de Lille, CNRS, INSERM, CHU de Lille, 59000 Lille, France; yves.rouille@ibl.cnrs.fr (Y.R.); jean.dubuisson@ibl.cnrs.fr (J.D.); sandrine.belouzard@ibl.cnrs.fr (S.B.)

**Keywords:** SARS-CoV-2, coronavirus, COVID-19, virus egress, virus release, transmission electron microscopy, serial sections

## Abstract

The mechanisms of severe acute respiratory syndrome coronavirus 2 (SARS-CoV-2) egress, similar to those of other coronaviruses, remain poorly understood. The virus buds in intracellular compartments and is therefore thought to be released by the biosynthetic secretory pathway. However, several studies have recently challenged this hypothesis. It has been suggested that coronaviruses, including SARS-CoV-2, use lysosomes for egress. In addition, a focused ion-beam scanning electron microscope (FIB/SEM) study suggested the existence of exit tunnels linking cellular compartments rich in viral particles to the extracellular space resembling those observed for the human immunodeficiency (HIV) in macrophages. Here, we analysed serial sections of Vero cells infected with SARS-CoV-2 by transmission electron microscopy (TEM). We found that SARS-CoV-2 was more likely to exit the cell in small secretory vesicles. Virus trafficking within the cells involves small vesicles, with each generally containing a single virus particle. These vesicles then fuse with the plasma membrane to release the virus into the extracellular space. This work sheds new light on the late stages of the SARS-CoV-2 infectious cycle of potential value for guiding the development of new antiviral strategies.

## 1. Introduction

The world is currently in the grip of a pandemic of coronavirus disease 19 (COVID-19), a new transmissible infectious disease caused by severe acute respiratory syndrome coronavirus 2 (SARS-CoV-2). An analysis of the genome of this virus showed it to be phylogenetically close to SARS-CoV-1, which emerged in 2003, and suggested that it probably originated from a bat coronavirus transmitted to humans [1]. The role of an intermediate animal species in this transmission is suspected, but the species concerned has not been clearly identified, and the conditions in which this interspecies transmission occurred are unknown. Nonetheless, the very rapid spread of this new infectious agent in the world population demonstrates its highly infectious properties. This virus has already infected more than 160 million people worldwide and caused more than 3.3 million deaths [2].

SARS-CoV-2 enters cells by fusing directly with the plasma membrane or by exploiting endocytosis mechanisms and then fusing with an endosomal membrane [3]. After fusion, the viral genome is released into the cytosol for translation and replication. Similar to other single-stranded positive-sense RNA viruses [4], coronaviruses modify host cell membranes to induce the formation of double-membrane vesicles (DMVs), in which intense viral RNA replication occurs [5]. The translation of the positive-sense sub-genomic viral RNA results in the synthesis of all viral proteins, including those necessary for the morphogenesis of progeny virions. These viral particles, similar to those of other coronaviruses, are predominantly spherical or ellipsoidal, with a mean diameter between 90 and 100 nm [6,7]. The main constituents of the viral particle are glycoprotein S, the transmembrane proteins M and E, and nucleoprotein N, which forms a viral ribonucleoprotein (vRNP) by complexing with the genomic viral RNA [8,9]. Viral assembly takes place on the cytoplasmic side of the intermediate compartment, between the endoplasmic reticulum (ER) and the Golgi apparatus (ERGIC), or in the Golgi apparatus [10].

As with other coronaviruses, the mechanism of viral egress is one of the least understood steps in the SARS-CoV-2 infectious cycle. Most enveloped RNA viruses are released by budding at the plasma membrane or by budding in internal host cell compartments; they then exit into the extracellular space via the biosynthetic secretory pathway. The budding of SARS-CoV-2 in the ERGIC compartment, or in the Golgi apparatus, suggests that this virus also exits the host cell via the secretory pathway. However, one recent study proposed a model in which coronaviruses, including SARS-CoV-2, use lysosomes for egress rather than the biosynthetic secretory pathway [11]. This study showed that brefeldin A (BFA), a fungal compound that disassembles the Golgi apparatus stacks, did not affect virions release. This unconventional release mechanism would require lysosome deacidification and the inactivation of lysosomal degradation enzymes to prevent the degradation of viral particles in this compartment [11]. Coronaviruses have previously been observed in lysosomal compartments on transmission electron microscopy (TEM), but the significance of these observations remained unexplored at the time [12]. However, it is unknown whether these intracellular compartments containing viral particles can fuse with the plasma membrane for virion release. A recent cryoFIB/SEM study (cryotechniques with a focused ion-beam scanning electron microscope) of SARS-CoV-2-infected cells suggested the presence of “exit tunnels”, linking virion-rich intracellular vacuoles to the extracellular space [13]. The authors suggested that SARS-CoV-2 could egress through these tunnels by a mechanism of exocytosis from these large intracellular cisterns.

Using conventional TEM, we recently described the ultrastructural characteristics of the complete infectious cycle of SARS-CoV-2 within the host cell [14]. In this new study, we added to these observations by analysing serial TEM sections of infected cells, focusing specifically on egress events. We show that SARS-CoV-2 viral particles seem to be released into the extracellular space principally via trafficking in small secretory vesicles, most of which contain a single viral particle. We did not observe any opening of intracellular virus-rich cisterns to the extracellular space. We conclude from these TEM observations that SARS-CoV-2 is probably released from its host cell in small secretory vesicles.

## 2. Materials and Methods

### 2.1. Cells and Virus

To generate virus stock, Vero-81 cells (kidney epithelial cells established from an African green monkey) were infected with the SARS-CoV-2 (strain BetaCoV/France/IDF0571/2020) as previously described [14]. Briefly, these cells (ATCC, CCL-81) were maintained in Dulbecco’s modified Eagles medium (DMEM) supplemented with 10% fetal bovine serum (FBS) at 37 °C under an atmosphere containing 5% CO2. SARS-CoV-2 infection was facilitated by transducing these cells with a lentiviral vector expressing transmembrane protease serine 2 (TMPRSS2). SARS-CoV-2 was then propagated in these Vero-81 cells expressing TMPRSS2. Ultrastructural analysis was then performed following infection with the produced viral particles on regular Vero-81 cells. Cells were infected at a multiplicity of infection (MOI) of 0.25 for 1 h and were fixed at various time points post-infection. Our previous kinetic study of these infected cells showed that the first massive viral exit events occurred at 10 h post-infection [14]. We therefore focused specifically on this time point in these investigations in order to avoid interference from possible concomitant viral re-entry and/or cell death events occurring at later points.

### 2.2. Electron Microscopy

Cells were fixed by incubation for 24 h in 4% paraformaldehyde and 1% glutaraldehyde (Sigma, St-Louis, MO, USA) in 0.1 M phosphate buffer (pH 7.2). They were then washed in phosphate-buffered saline (PBS) and post-fixed by incubation for 1 h with 2% osmium tetroxide (Agar Scientific, Stansted, UK) in 0.15 M phosphate buffer. They were dehydrated in a graded series of ethanol solutions and propylene oxide, impregnated with a mixture of (1:1) propylene oxide/Epon resin (Sigma), and then embedded in pure Epon resin, which was allowed to polymerise for 48 h at 60 °C. The Epon blocks were precisely resized for the cutting of a ribbon of serial ultrathin sections (80 nm thick) of individual cells with a Leica Ultracut UCT ultramicrotome (Leica Microsysteme GmbH, Wien, Austria). These serial sections were then placed on TEM nickel one-slot grids (Agar Scientific, Ltd., Stansted, UK) coated with Formvar film. The resulting TEM grids were stained for 20 min with 2% uranyl acetate (Merck, Darmstadt, Germany) and 5% Reynolds lead citrate for observation with a Jeol 1011 (Jeol Ltd., Tokyo, Japan) electron microscope. Electron micrographs of the sections were recorded with a digital camera driven by Digital Micrograph software (GMS 3, Gatan, Pleasanton, CA, USA) for the same region of cells in each of the series of sections.

## 3. Results

We investigated eight cells in which numerous viral particles were observed intracellularly and at the surface by serial section imaging. For five cells, we investigated two or three different areas, resulting in total section imaging for 16 different subcellular domains. For clarity, Table 1 below lists the serial sections corresponding to the cell regions shown in the manuscript figures and those shown in the Appendix A.

Figure 1 shows a typical cell and the three types of cellular organisations frequently encountered in the areas of virus release into the extracellular space: (i) areas in which the viral particles exit via a cavity formed by a large cytoplasm invagination and numerous microvilli of the plasma membrane (part 1); (ii) areas in which the viral particles exit at a portion of the plasma membrane without any particular protrusion (part 2); (iii) intermediate situations, in which the viral particles exit in areas displaying microvilli at the plasma membrane (part 3). For this cell (cell A on Figure 1), we examined each of these areas carefully through an analysis of serial sections: Figure 2 for part 1 (nine serial sections), Figure 3 for part 2 (six serial sections), and Figure 4 for part 3, (six serial sections). In Figure 2, Figure 3 and Figure 4, large intracellular vesicles rich in viral particles (thin black arrows) and small vesicles each containing only one viral particle (large white arrows) were recorded. Comparisons of these observations of individual sections with observations of the preceding or following serial sections did not demonstrate the presence of tunnels connecting the plasma membrane to intracellular vesicles containing numerous viral particles in any of the three virus release situations shown in Figure 1. Particular attention was paid to the situation shown in part 1 of cell A (Figure 2), as the long microvilli of the plasma membrane, often surrounded by viral particles, might potentially resemble “exit tunnels”. However, a meticulous analysis of the serial sections showed that these structures were simple microvilli of the plasma membrane and not tunnels.

The absence of tunnel was confirmed by examining several other cells presenting the same feature. Similar observations were made for cell B, presented in Figure 5 and Figure 6. Figure 5 shows cell B at low magnification, and the area of this cell further studied by serial sections in Figure 6. This particular area presents a characteristic situation, with a cavity formed by a large cytoplasmic invagination accompanied by numerous microvilli of the plasma membrane and the active release of viral particles. Figure 6 shows an analysis of 12 serial sections in this particular region. A complete series of 24 consecutive electron micrographs of this region can be consulted in the Appendix A. As for the previous figures, large intracellular vesicles rich in viral particles (thin black arrows) and small vesicles each containing only one viral particle (large white arrows) were recorded (Figure 6). Again, comparisons of an individual section with the preceding and following serial sections suggested that the virions were trafficked towards the plasma membrane in small vesicles. The structure identified in brackets from sections 4 to 12, which is suggestive of a tunnel, was found to be a simple invagination of the plasma membrane not connected to an intracellular compartment. Interestingly, our observations suggest that virions were released into the thin extracellular space formed by this invagination via small secretory vesicles, each containing a single virus particle (thin white boxes on sections 10 and 11, also shown in the inset on these sections). On section 10, another similar virus release event can be observed (thin white arrow in the upper part of the image, also shown in the inset).

## 4. Discussion

Viral particle egress is one of the least well understood steps in the infectious cycle of SARS-CoV-2 in its host cell. To date, few data have been published on this topic for this and other coronaviruses. Viral morphogenesis is known to involve budding in intracellular compartments, so viral particles are generally thought to be released via the biosynthetic secretory pathway. However, one recent study suggested that coronaviruses, including, potentially, SARS-CoV-2, make use of lysosomal trafficking for egress rather than the common secretory pathway [11]. It has also recently been shown that flaviviruses (dengue and Zika viruses) use autophagosome-derived organelles for egress [15]. However, it is unknown whether these intracellular compartments containing coronaviruses or flaviviruses can fuse with the plasma membrane to release the large numbers of viral particles present in these compartments.

A recent cryoFIB/SEM imaging study of Vero cells infected with SARS-CoV-2 has suggested the presence of “exit tunnels” connecting large intracellular virus-containing vesicles to the plasma membrane [13]. The authors suggested that these tunnels resulted from the fusion of these large virus-containing vesicles with the cell membrane, allowing the release of numerous SARS-CoV-2 viral particles through an exocytosis-like mechanism. Such a mechanism is conceivable, as a similar mechanism has previously been described for human immunodeficiency virus (HIV) in macrophages, in which HIV buds in dilated intracellular compartments [16,17]. Many of these intracellular compartments have been shown to be connected to the plasma membrane and have thus been named intracellular plasma membrane-connected compartments (IPMCs) [18]. Three-dimensional electron microscopy analyses have suggested that IPMCs represent portals through which HIV may be released into the extracellular space [17]. However, in the case of macrophages, similar compartments were previously found to exist in uninfected cells [17].

Our analysis of SARS-CoV-2-infected Vero cells based on serial TEM sections provided no evidence for the presence of such exit tunnels, as shown in the serial sections of the 4 cell subdomains presented in our figures but also in the 12 other cell subdomains analysed in 3D from 6 other different cells. On the contrary, our analysis suggested that the viral particles were individually trafficked in small secretory vesicles. Some of these secretion vesicles were presumably opened to the extracellular space, leading to a possible release of the virus (Figure 6, insets in sections 10 and 11). Viral particles were often released in areas of cytoplasmic invagination containing many microvilli of the plasma membrane. Similar observations have been reported in other recent studies of Vero cells infected with SARS-CoV-2 that were examined by scanning electron microscopy (SEM) [19] or coupled SEM and TEM [19,20]. The arrangement of parallel microvilli may resemble tunnels, but our observations show that they are never connected to virus-rich intracellular vesicles. Moreover, the virions can be released into very thin extracellular spaces between these microvilli. Further studies will be required to determine whether this situation enables the virus to hide from the immune system.

Further investigations, including functional studies, will also be necessary to determine the origin of these small secretory vesicles transporting the virions and the pathway used for their progression towards the plasma membrane. Early coronavirus studies performed with the mouse hepatitis virus model suggested that after budding into the ERGIC, the virus follows the classical secretory route to the cell surface [21,22]. However, as the coronaviruses egress has been recently shown to be BFA resistant [11], these vesicles may not belong to the classical secretion pathway and rather use an unconventional secretory pathway. The vesicles observed fusing with the cell surface to release the virus in the extracellular space may have different origins and use different pathways that by-passes the Golgi stacks, as recently suggested [23]. They could follow a pathway through the Golgi apparatus deviating to the lysosomes and/or a pathway from the non-compact zones of the Golgi ribbon constituted by the ERGIC and recycling endosomes [23].

The cryoFIB/SEM study cited above also reported the occurrence of plasma membrane discontinuities next to virus particles outside the cell, suggesting possible viral release through membrane lesions [13]. We observed no such discontinuities, with our electron micrographs instead suggesting viral release without plasma membrane rupture via the possible exocytosis of small vesicles, most of which contained a single viral particle. The reasons for these differences between these two studies are unknown but may be related to the timing of the analysis (10 h post-infection in our serial section study versus 24 h post-infection in the cryoFIB/SEM study). It is possible that cell lysis contributes to virion release into the extracellular space at 24 h but not 10 h post-infection. Further studies at different times post-infection will probably provide clearer insight into the various possible mechanisms of viral exit.

Nevertheless, our study suggests that small secretory vesicles carrying a single viral particle represent the main means of egress for SARS-CoV-2, at least at the early stage of infection. We cannot rule out the possibility that egress events involving large vesicles have been missed due to the limited number of cells analysed. However, these cells were fully analysed in 3D using this serial section approach, and none of them showed an egress event involving an opening of large intracellular vesicles to the outside, directly or via an exit channel. It will be certainly important to continue this type of investigation with other, more relevant, cellular models, such as human primary airway epithelial cells, although these cells will most certainly be more difficult to handle for a serial TEM sections approach. This work sheds new light on late stages in the SARS-CoV-2 infectious cycle, which could be useful for the development of new antiviral strategies in a context in which few therapeutic approaches are currently available.

## Figures and Tables

**Figure 1 cells-10-02047-f001:**
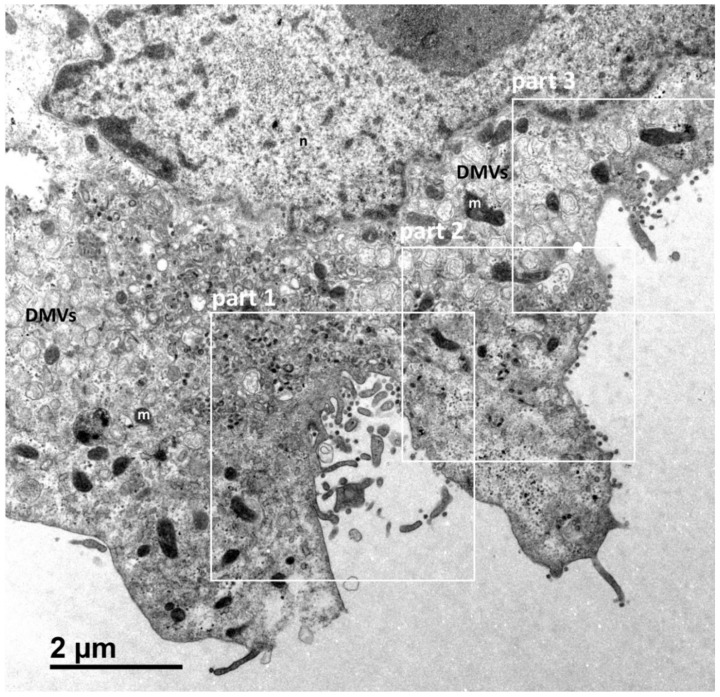
Vero cell at 10 h post-infection with SARS-CoV-2. This cell (cell A) is representative and shows the various virus release situations observed. The three areas delimited by white squares designate the regions of the cell analysed by the examination of serial sections in subsequent Figure 2, Figure 3 and Figure 4: a region characterised by a large cytoplasm invagination rich in plasma membrane microvilli (part 1); a region in which the virus is released and the plasma membrane does not exhibit particular protrusion (part 2); and a region in which the situation is intermediate, with small plasma membrane microvilli at the cell surface (part 3). The cellular area between the cytoplasm invagination and the nucleus (n) contains a large number of viral particles. This area is flanked on both sides by numerous double-membrane vesicles (DMVs) and mitochondria (m).

**Figure 2 cells-10-02047-f002:**
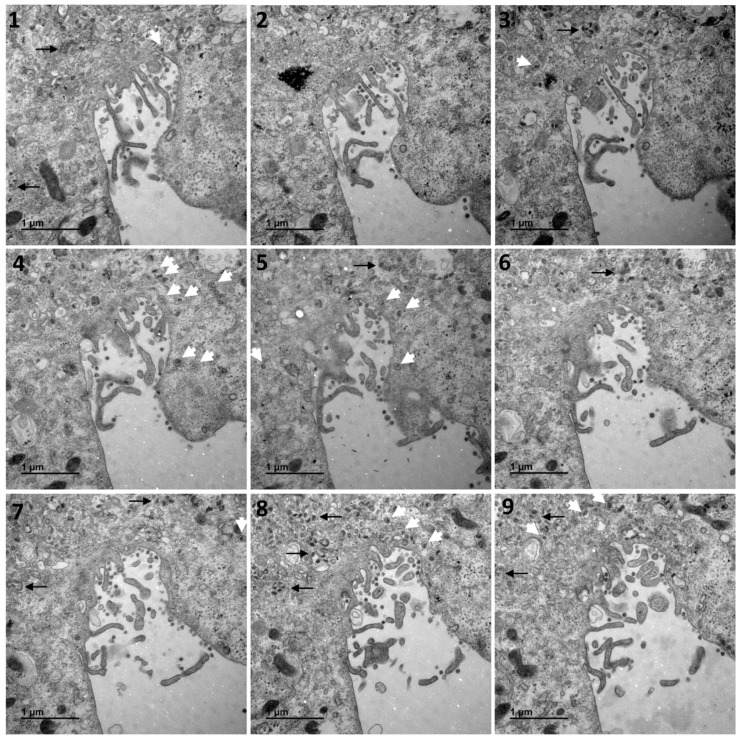
Serial TEM sections of cell A part 1, a region characterised by a large cytoplasm invagination rich in plasma membrane microvilli. Large intracellular vesicles rich in viral particles (thin black arrows) and small vesicles each con-taining a single viral particle (large white arrows) are shown on each serial section. These electron micrographs of a series of 9 TEM sections (**1**–**9**) are part of a larger series (11 sections) provided at high resolution in the Appendix A. The numbers on the panels indicate the successive serial TEM sections.

**Figure 3 cells-10-02047-f003:**
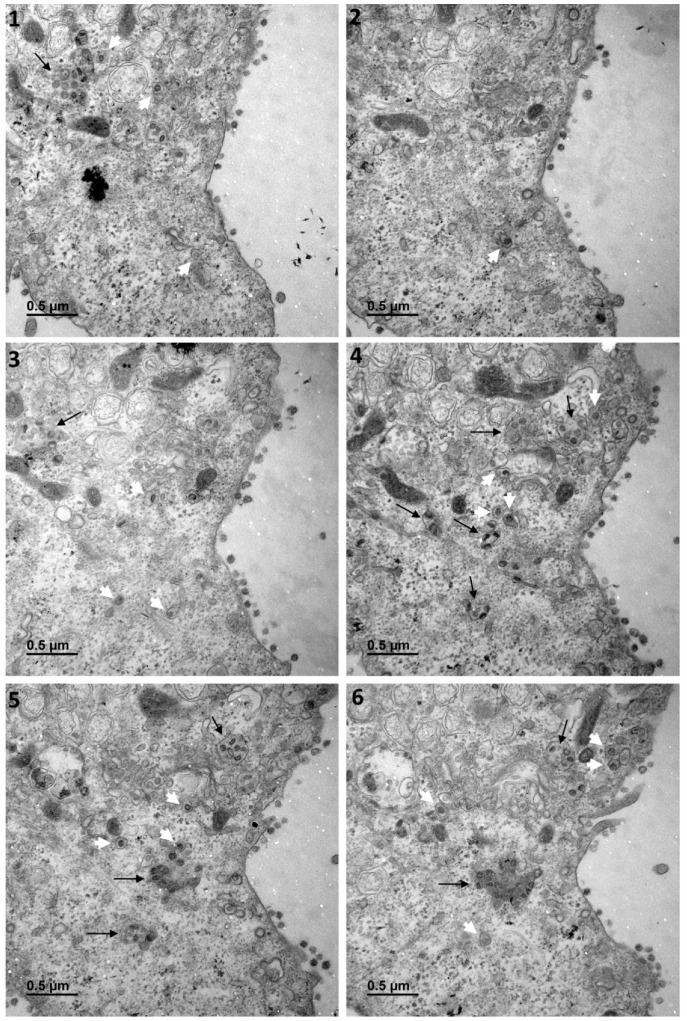
Serial TEM sections of cell A part 2, a region in which the virus is released at the plasma membrane without any particular protrusion. Large intracellular vesicles rich in viral particles (thin black arrows) and small vesicles each con-taining a single viral particle (large white arrows) are shown on each serial section. These electron micrographs of a series of 6 TEM sections (**1**–**6**) are part of a larger series (7 sections) provided at high resolution in the Appendix A. The numbers on the panels indicate the successive serial TEM sections.

**Figure 4 cells-10-02047-f004:**
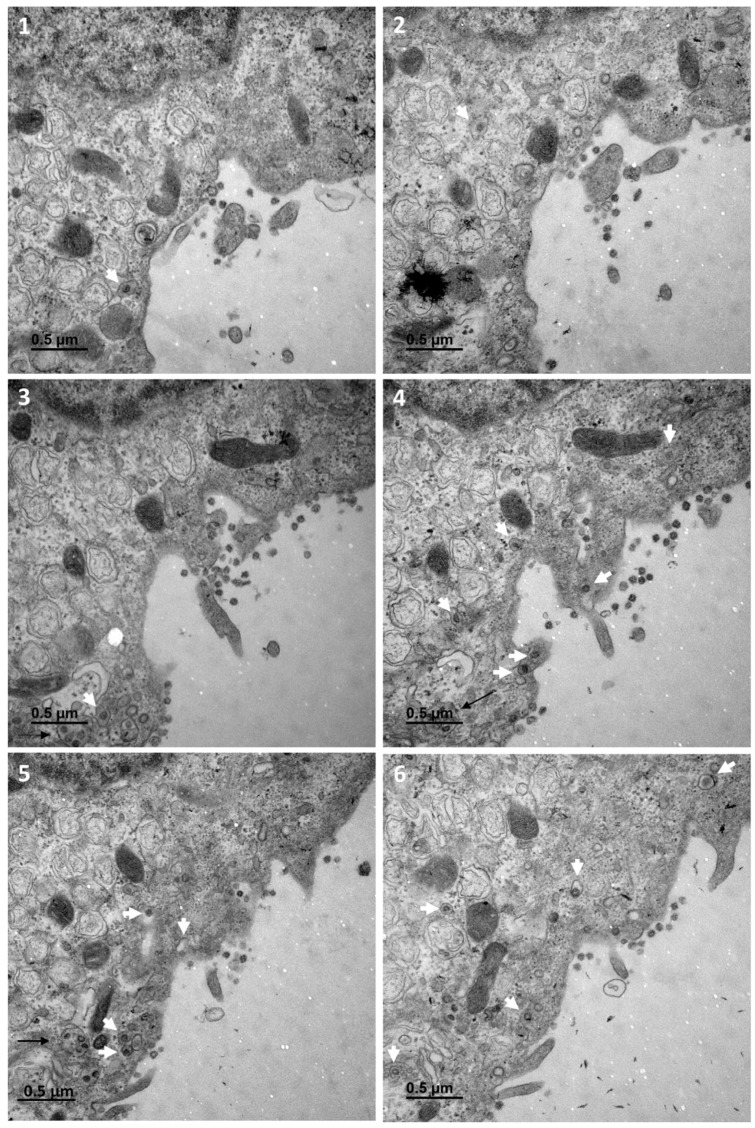
Serial TEM sections of cell A part 3, a region in which the situation is intermediate to those observed on part 1 and 2, with small plasma membrane microvilli at the cell surface. Large intracellular vesicles rich in viral particles (thin black arrows) and small vesicles each containing a single viral particle (large white arrows) are shown on each serial section. The high-resolution electron micrographs for this series can be consulted in the Appendix A. The numbers on the panels (**1**–**6**) indicate the successive serial TEM sections.

**Figure 5 cells-10-02047-f005:**
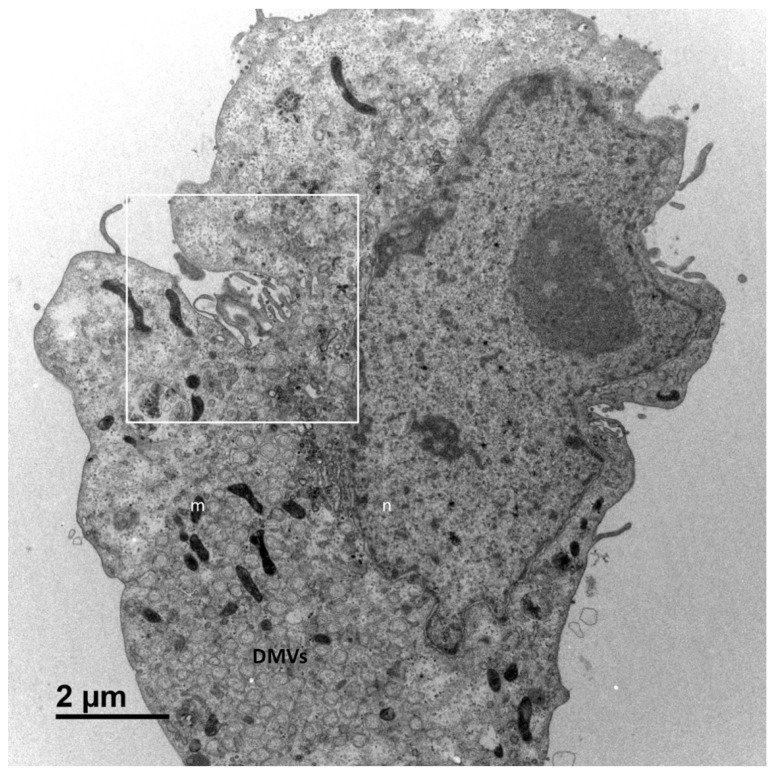
Vero cell at 10 h post-infection with SARS-CoV-2. This cell (cell B) is representative of a frequently observed viral release situation characterised by a large cytoplasm invagination rich in plasma membrane microvilli (area designated by the white square, analysed by the examination of serial TEM sections in Figure 6). The cellular area between the cytoplasm invagination and the nucleus (n) contains a large number of viral particles. This area is close to numerous double-membrane vesicles (DMVs) and mitochondria (m).

**Figure 6 cells-10-02047-f006:**
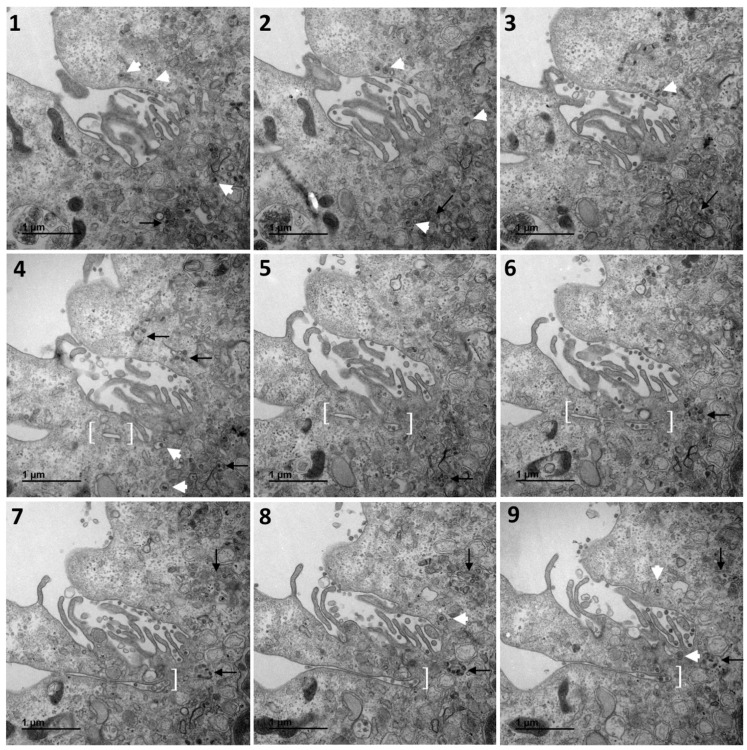
Serial TEM sections of the previous area of cell B, characterised by a large cytoplasm invagination rich in plasma membrane microvilli. Large intracellular vesicles rich in viral particles (thin black arrows) and small vesicles each con-taining a single viral particle (large white arrows) are shown on each serial section. The structure identified in white brackets from sections 4 to 12 appears to be an invagination of the plasma membrane, forming a very thin extracellular space between two cellular microvilli. These electron micrographs of a series of 12 TEM sections are part of a larger series (24 sections) provided at high resolution in the Appendix A. Viral particles are released into the thin extra-cellular space between two cellular microvilli through small secretory vesicles each containing a single virus particle (thin white boxes in sections 10 and 11, also shown in insets). On section 10, another viral release event involving a small se-cretory vesicle containing a single viral particle can be visualised (thin white box in the upper part of the image, also shown in the inset). The numbers on the panels (**1**–**15**) indicate the successive serial TEM sections.

**Table 1 cells-10-02047-t001:** List of the different cells, cell regions, and corresponding serial sections analysed in this study.

Cell	Region	Figure	Appendix A (High Resolution)
Cell A	Whole cell	Figure 1	
	Part 1	Figure 2 (sections 1 to 9)	Sections 1 to 11
	Part 2	Figure 3 (sections 1 to 6)	Sections 1 to 7
	Part 3	Figure 4 (sections 1 to 6)	Sections 1 to 6
Cell B	Whole cell	Figure 5	
	Part 1	Figure 6 (sections 1 to 15)	Sections 7 to 16

## Data Availability

All the electron micrographs of the serial sections of the two cells fully analysed in this manuscript, and additional serial sections in the continuity of these series, are provided in the Appendix A. Electron micrographs of the serial sections of the other six cells analysed in this study (12 different subcellular domains) can be provided at high resolution upon request to the corresponding author.

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
