# Peer review of "Secretory Vesicles Are the Principal Means of SARS-CoV-2 Egress"

_cells, 2021, doi:10.3390/cells10082047_

Round 1

Reviewer 1 Report

I think this 2nd round of revision, where the discussion of potential pathways of vesicle egress has been expanded, makes the manuscript acceptable for publication in Cells.

Author Response

We would like to thank Reviewer #1 for this positive view of our work and the constructive remarks made during the two rounds of revision.

Reviewer 2 Report

Eymieux et al. reported SARS-CoV-2 was released through small secretion vesicles and most of the vesicles contains only one particle by TEM. Although this study described the details of SARS-CoV-2 release, the evidence is not solid. The whole study is quite preliminary and more evidence from other mythologies are required.

  1. Line 186. The authors concluded that most small vesicles only contains one viral particle. Statistics analysis is required to make this conclusion using multiple fields.
  2. All of the analysis was only based on Vero cells which is not a physiological cell line. The budding process may be quite different in lung cells compared with monkey kidney cells. Therefore, it will be necessary to confirm the findings in lung cell lines. It will be better if the evidence from primary airway epithelial cells can be provided.
  3. Other lines of evidence should be provided to confirm the findings. Functional validations will be necessary to support the key conclusions.
  4. Considering only limit number of cells were analyzed, the conclusion is premature. Enough cells and fields as well as statistics analysis should be included in this study.

Author Response

Eymieux et al. reported SARS-CoV-2 was released through small secretion vesicles and most of the vesicles contains only one particle by TEM. Although this study described the details of SARS-CoV-2 release, the evidence is not solid. The whole study is quite preliminary and more evidence from other mythologies are required.

  1. Line 186. The authors concluded that most small vesicles only contains one viral particle. Statistics analysis is required to make this conclusion using multiple fields.

To appreciate our approach, it is very important to understand that the visualization of virus exit events relies on an analysis of serial electron microscopy sections. Thus, a small virus-containing intracellular vesicle that is only visualized on a single 80 nm section (neither on the previous section nor on the next section) cannot contain more than one virion.

  1. All of the analysis was only based on Vero cells which is not a physiological cell line. The budding process may be quite different in lung cells compared with monkey kidney cells. Therefore, it will be necessary to confirm the findings in lung cell lines. It will be better if the evidence from primary airway epithelial cells can be provided.

We agree with this remark, although the realization of serial sections will be much more difficult to realize with this type of cells that are cultivated in monolayers and are thus much more fragile. This is now discussed in the revised version of our manuscript :

“It will be certainly important to continue this type of investigation with other, more relevant cellular models such as human primary airway epithelial cells, although these cells will most certainly be more difficult to handle for a serial TEM sections approach.” Lines 277 to 279 in the revised manuscript.

  1. Other lines of evidence should be provided to confirm the findings. Functional validations will be necessary to support the key conclusions.

We agree with this remark. We have modified one sentence in our discussion to include this notion of functional studies in the revised version of our manuscript :

“Further investigations, including functional studies, will also be necessary to determine the origin of these small secretory vesicles transporting the virions and the pathway used for their progression towards the plasma membrane.” Line 248 in the revised manuscript.

  1. Considering only limit number of cells were analyzed, the conclusion is premature. Enough cells and fields as well as statistics analysis should be included in this study.

It is true that only a small number of cells were analyzed but these cells were analyzed in their complete volume, by this approach of serial TEM sections which requires a meticulous and extremely laborious work. At the end, several hundred sections were observed in this study.

Round 2

Reviewer 2 Report

I appreciate the response from the authors to address my concerns. Even though it will be difficult to capture more qualified fields with virions for further analysis, this part is essential and it will be the fundament for the whole story. This work will be important only if the evidence is solid enough. I highly recommend to add more solid evidence for this part.

Author Response

Our response:

Our study is of course morphological and it is impossible to show all the observations we made, which concern hundreds of sections. As mentioned on lines 119-121, we analyzed 8 different cells and several different virus-releasing subcellular domains of these cells, thus representing 16 different subcellular domains that were fully analyzed in 3D. It was of course not possible to show everything in the manuscript, in which we present on our figures the analysis of 4 different subcellular domains. But the observations made on these 16 different subcellular domains were exactly the same as those shown on our figures. In order to clarify this information, which may not have been visible in our manuscript, a sentence has been modified on lines 235-237 :

“Our analysis of SARS-CoV-2-infected Vero cells based on serial TEM sections provided no evidence for the presence of such exit tunnels, as shown in the serial sections of the 4 cell subdomains presented in our figures but also in the 12 other cell subdomains analyzed in 3D, from 6 other different cells.”

This manuscript is a resubmission of an earlier submission. The following is a list of the peer review reports and author responses from that submission.

Round 1

Reviewer 1 Report

In Eymieux et al. study, the authors analysed serial sections of Vero cells infected with SARS-CoV-2 by transmission electron microscopy (TEM). They found that SARS-CoV-2 was more likely to exit the cell in small secretion vesicles, each generally containing a single virus particle. They also found that these vesicles fuse with the plasma membrane to release the virus into the extracellular space.

Although we consider that this study is a solid research work, we ask for minor modifications that will undoubtedly add value to Eymieux’s work. You will find below my remarks and recommandations.

Generally, much attention should be paid by the authors regarding their conclusions. If we agree that their images suggest that SARS-CoV-2 exit the cells in small secretion vesicles, each generally containing a single virus particle, their images are not convincing enough for supporting that these vesicles fuse with the plasma membrane to release the virus into the extracellular space. Supplemental images or better figures explanation/legends are needed. My main concern is thus not regarding the results of the authors, but rather their conclusions. Attention has to be paid for elaborating hypothesis instead. This point will not decrease the impact of authors study but will open for alternative interpretations.

Here are my points of concern:

A table listing the cells and the number of regions and serial sections analyzed should be provided by the authors. Authors might consider also adding some numbers regarding the single-virus containing vesicles versus several virus containing vesicles to support their claims.

Lines 103-106: as information, authors might consider for future studies the use of 2.5 % glutaraldehyde and of potassium-hexacyanoferrate/ 1% osmium rather than a mix of paraformaldehyde and glutaraldehyde for fixation and phosphate buffer.

Lines 123-128 : The description of the three areas examined should be rewritten. First ‘three types of situations” should be replaced by ‘cellular organization’ or any more scientific terms. Secondly, we do not think that ‘flat’ is an adequate term for part2 region. Reference to precise cellular features should rather be employed, as for example the presence/absence of microvilli on the plasma membrane, the content of the cytoplasm in terms of vesicles, mitochondria or any other cellular compartments. Precision is needed.

Figures 1-3: thin black arrows should be enlarged to better visualization (or large black arrows versus large white arrows would be better).

Line 133-135: maybe this sentence is more intended for discussion.

Lines 138-142: ‘protrusions’ may be changed for ‘microvilli’, here and elsewhere in the manuscript if referring to the same structures.

Lines 167-168: which finding? which same feature? Be more precise.

Figure 6: dashed boxes with numbers should be used instead of small white arrows for pointing at zoomed insets, otherwise there might be confusions. As it is, not easy to read the figure.

Lines 202-205, Figure6 legend: please correct this, probably a mistake.

Lines 181-183: based on ultrastructure only, virus release might be suggested rather than assumed.

Line 227: 3D electron microscopy should be precised

Line 231: “This analyses..”: please replace ‘this’ by ‘our’ or ‘the present’ study, otherwise confusing.

Line 233: ‘suggested’ maybe rather than ‘indicated’. Again one can only assume a vesicle release based on ultrastructure only. Time-lapse videomicroscopy analysis, maybe combined to CLEM would be needed for a definitive assumption. Maybe for discussion an opening for continuing your work might be of interest.

Lines 234-235: this sentence would need an illustration. Otherwise not convincing. If present in the figures please refer to it.

Lines 235-236: Please explain. On what basis do you assume that: amount of extracellular particles, amount of intracellular particles, release configurations…????? Be more precise. Discriminate between amount of particles and release notion. Particles might be concentrated in a region after a diffuse release…  

Lines 242-243: very interesting, the idea may be expanded somehow for more clear meaning.

Line 244: ‘transporting’ rather than ‘releasing’ might be considered.

Line 245: please add reference for BFA-treated cells, even if already mentioned in introduction.

Lines 250-254: 1) we did not see clear images of exocytosis along the manuscript. Precaution should be used regarding authors conclusions. Hypothesis should be elaborated rather than complete conclusions. Additional work would be needed for definitive conclusion about Sars-CoV-2 release through exocytosis of single-virus containing vesicles. Reference to membrane lesions from study (13) might be discarded as it does not bring additional value to the study message. 

Lines 262 and 267: these two sentences might be contradictory.  I would suggest considering late-stages of Sars-CoV-2 infectious cycle rather when cell death occurs, more after 18-24hpi. So I would advice the authors to refer to early-stage of infection in the present study for 10hpi.

In discussion section, authors might consider discussing the differences in the methodology used by the group cited in (13) and by the authors: group (13) performed 3D EM reconstructions of cells adherent to a 2D-culture substrate, while authors performed cuts through a pellet of cells. This difference might be of importance for explaining differences between the two studies.

Author Response

Please see the pdf enclosed.

Reviewer 2 Report

The submitted manuscript is a follow-up of a previous more comprehensive TEM analysis by these authors (Eymieux et al. Cell. Mol. Life Sci. 2021), where Vero cells infected with SARS-CoV-2 were studied ultrastructurally with TEM and IF 6, 8, 10, 12, and 24 hrs after infection. In the present manuscript the cells are studied 10 hrs after infection by TEM, only (but using serial sections), which was the earliest time point in the previous study when virus particles were observed leaving the host cells. Looking at the process at a relatively early time point is an advantage, since the cellular machinery for virus egress is more intact than at later time points.

The manuscript could be suitable as a Short Communication, but I am not sure whether Cells has this category.

The main conclusion seems to be that SARS-CoV-2 particles reach the cell surface individually in small vesicles and not via “tunnels” as proposed by Zhang et al. (ref 13).

The manuscript is methodologically limited, although the serial section analysis is informative. No markers are included of secretory or other endomembrane compartments either in the LM of EM level.

Small vesicles and larger pleomorphic virus-containing structures are assigned possible functions mainly based on their spatial properties. However, both can be seen at the cell periphery. Both can be dynamic. In their previous publication the authors state that both types are involved in virus egress. Overall, the conclusion that the small vesicles represent the terminal secretory carriers is not very convincing. Good images showing exocytosis of the small vesicles are not included here. In fact, there are better images in their earlier paper - but, the large virus-containing carriers also fuse with the PM. It is difficult to observe a significant number of events where fusion of small vesicles with the PM is ongoing. Such images should be provided at higher resolution - as in their previous publication on the virus life cycle after infection.

The authors do not discuss from where the vesicles they observe could be derived and how they move towards the PM. If these are individual vesicles - where did they acquire their membrane? Such a discussion would be required.

Author Response

Please see the pdf enclosed.

Reviewer 3 Report

In this manuscript, the authors extend their earlier ultrastructural analysis of SARS-CoV-2 replication in Vero cells, a commonly used model cell line for SARS-CoV-2 studies (Eymieux et al. Cell. Mol. Life Sci. 2021).  They now focus on the late stages of the SARS-CoV-2 replication cycle and release by examining serial 80 nm ultrathin section of plastic embedded cells 10 hrs after infection.  Medium magnification images are presented from selected areas of two cells.  Given the size of SARS-CoV-2 (80-90 nm, i.e. similar to the section thickness) this is a challenge.  Furthermore, with the use of plastic-embedded material, any interpretation of the images relies on morphological features, and the images can only represent a snapshot of the complex virus release processes in virus-producing cells.

These studies cry out for segmentation and 3D reconstruction! What are the criteria for identification of the SARS-CoV-2 particles - vs. size, electron density, and round shape (after all spikes cannot be seen at this magnification).  Tracing out membranes and highlighting presumptive virus particles in colour would show how these structures were identified/which structures are considered to represent viruses, and tracing membranes would clarify how virus-containing membranes are connected in 3D, and whether vesicles with a single virus particle are not in fact connected in 3D.  For example:

  • Fig 2 section 4, with seven 7 white arrows indicating small vesicles each containing a single virus particle. However, the ‘virus’ structures inside these vesicles appear different from the extracellular SARS-CoV-2 particles, with several having an elongated kidney-shaped morphology. In section 5: the second white arrow points to a vesicle containing one structure that is larger than a SARS-Cov-2 particle and one which is significantly smaller.
  • Fig 2 section 3 black arrow: What is the evidence that the three dark structures are SARS-CoV-2 particles?  They are darker than the other virus particles in the image, not round, and the long dimension is greater than the diameter of the SARS-Cov-2 particles at the cell surface.
  • Fig 2 section 8, top left of the photo (with black arrows): All the presumptive virus particles in this region appear larger, darker and more elongated than the extracellular SARS-CoV-2 particles. 
  • Can you confirm that these are adjacent 80 nm sections – it is hard to understand why the larger vesicles with multiple particles are only apparent in one section and cannot be traced in the adjacent section.
  • Figure 4 – the structures in section 3 and 4 are in fact vaguely similar to the tunnels that have been described. Please do not obscure important features with scale markers (Fig. 4 panel 4).  The scale marker could be placed bottom right in the extracellular space.  I presume all images are shown at the same magnification, in which case a single bar would suffice for each figure.
  • Segmentation (tracing of membranes and identifying all virus particles) is particularly pertinent for the interesting structure in Cell B (Figure 6). Please include sections 13 – 16 in the main article (where the narrow fold or “tunnel” connects to the virus-filled invagination)
  • Regarding images of virus release from small vesicles – can you exclude that the structures shown in Fig. 6 Section 10 left inset and in Section 12 are not clathrin-coated pits (i.e. endocytic structures)?

Given that fusion figures of vesicles with the cell surface leading to virus release are likely to be short-lived transient structures and that no specific markers for such exocytic events are available, the statement that ‘small secretion vesicles carrying a single viral particle represent the main means of egress for SARS-CoV-2’ cannot be justified.

Author Response

Please see the pdf enclosed.

Round 2

Reviewer 2 Report

Please, see attached file. 
